# Synthesis and Characterization of Curcumin-Chitosan Loaded Gold Nanoparticles by *Oryctes rhinoceros’* Chitin for Cosmeceutical Application

**DOI:** 10.3390/molecules28041799

**Published:** 2023-02-14

**Authors:** Nurul Alyani Zainol Abidin, Faridah Kormin, Nurul Akhma Zainol Abidin, Mohd Fadzelly Abu Bakar, Iqbal Ahmed Moujdin

**Affiliations:** 1Faculty of Applied Sciences and Technology, Universiti Tun Hussein Onn Malaysia, Pagoh Education Hub, Pagoh 84600, Johor, Malaysia; 2Center of Excellence in Desalination Technology, King Abdulaziz University, P.O. Box 80200, Jeddah 21589, Saudi Arabia; 3Department of Mechanical Engineering, King AbdulAziz University, P.O. Box 80200, Jeddah 21589, Saudi Arabia

**Keywords:** curcumin, cosmeceutical nanotechnology, chitosan nanoparticle, rhinoceros beetle’s chitosan, gold nanoparticle, *Oryctes rhinoceros*

## Abstract

A breakthrough in cosmeceuticals by utilizing insects as major ingredients in cosmetic products is gaining popularity. Therefore, the interest in rare sources of ingredients, for instance, from the *Oryctes rhinoceros* beetle, can bring huge benefits in terms of turning pests into wealth. In this study, curcumin was chosen as the active ingredient loaded into chitosan-gold nanoparticles (CCG-NP). Curcumin is unstable and has poor absorption, a high rate of metabolism, and high sensitivity to light. These are all factors that contribute to the low bioavailability of any substance to reach the target cells. Therefore, chitosan extracted from *O. rhinoceros* could be used as a drug carrier to overcome these limitations. In order to overcome these limitations, CCG-NPs were synthesized and characterized. Chitosan was isolated from *O. rhinoceros* and CCG-NPs were successfully synthesized at 70 °C for 60 min under optimal conditions of a reactant ratio of 2:0.5 (0.5 mM HAuCl_4_: 0.1% curcumin). Characterizations of CCG-NP involved FTIR analysis, zeta potential, morphological properties determination by FE-SEM, particle size analysis, crystallinity study by XRD, and elemental analysis by EDX. The shape of the CCG-NP was round, its size was 128.27 d.nm, and the value of the zeta potential was 20.2 ± 3.81 mV. The IC50 value for cell viability is 58%, indicating a mild toxicity trait. To conclude, CCG-NP is a stable, spherical, nano-sized, non-toxic, and homogeneous solution.

## 1. Introduction

Cosmeceuticals are products with both aesthetic and therapeutic properties that are meant to improve both skin health and appearance merge in one application. They are applied topically as creams or lotions, similar to cosmetics, but include active ingredients that influence skin cell function. In order to make it function more effectively, the size of cosmeceuticals needs to be nanosized. Nanocosmeceuticals have a variety of advantages. There are many factors that can be used to control how active compounds are released from carriers. These factors include physical or chemical interactions between components, drug composition, polymer and additives, ratio, and the process of making them [1].

The cosmetics industry has achieved a breakthrough in the use of insects as vital components in its products in recent years [2,3]. Over the last 50 to 60 years, the cosmetic industry has developed from an era of secret formulations, elusive promises, and false optimism to a totally new sector founded on science. Cosmetics are no longer a stand-alone enterprise [4]. They are becoming more reliant on the cosmetics, pharmaceutical, biochemical, and medical industries [4]. As a result, new research productively turns into more potent cures and preventative measures. To achieve optimum effectiveness and safety, active ingredients in new cosmetic products must be carefully chosen. As a result, novel cosmetics have grown more complex in terms of formulations and presentation, such as medication administration and carrier for improved bioavailability of bioactive chemicals on target cells [5].

Recently, drug carriers that enhance the topical action of active compounds have gotten a lot of attention, and chitin is one of them [6,7]. After cellulose, chitin is the second most prevalent natural biopolymer. It is a key component of many fungi’s cell walls, insect exoskeletons, and crustacean shells. It is mostly made up of waste from the processing of marine foods such as crab, shrimp, and krill shells [8,9,10]. For commercial chitin production, crustacean sources have been favored [7,11,12]. Therefore, non-conventional chitin and chitosan sources such as corals, fungi, and insects might need to be explored to increase industry confidence in using them as an alternative to commercial chitin and chitosan.

The biologically active component of turmeric, rhizomes of *Curcuma longa* that belong to the ginger family, Zingiberaceae is the phenolic compound curcumin also known as diferuloylmethane; [1,7-bis-(4-hydroxy-3-methoxyphenyl)-1,6-heptadiene-3,5-dione] [13]. Chalcones are a subclass of flavonoids [14] with a 1, 3-diphenyl-2-propen-1-one skeleton [15]. In fact, analogous to curcumin, the structure of chalcone consists of two aromatic rings linked by a three-carbon α,β-unsaturated carbonyl system. [15,16]. Compounds containing α,β-unsaturated carbonyl-based moieties are often reactive. The reactivity of these groups is responsible for their diverse pharmacological activities [16].

Since it is substantial to have an effective cosmeceutical product, incorporating curcumin into cosmetics formulation will certainly turn it into a therapeutic product. Curcumin is an active compound with many benefits. Curcumin has been utilized in anti-aging, anti-acne, and skin-whitening products. However, one major issue with using this ingredient in pharmaceutical and cosmetic formulations is that it rapidly degrades when exposed to high temperatures and light. Curcumin’s bioavailability is limited by its low absorption, rapid metabolic breakdown, and rapid systemic clearance, limiting its vast therapeutic potential [17]. To achieve the goal of cosmeceuticals in dermatology research, it is important to produce new ecologically friendly nanosized compounds for skin nanoparticulate systems. However, curcumin has low intrinsic activity, poor absorption, a high rate of metabolism, inactivity of metabolic products, and quick excretion and clearance from the body. These are all factors that contribute to the low bioavailability of any substance to reach the target cells. Curcumin’s bioavailability and biological activity are currently being studied due to its low systemic bioavailability and limited access to certain tissues at sufficient pharmacologic levels in vivo. Several delivery strategies are being tested to improve curcumin’s bioavailability and biological activity [18].

The active application of AuNPs centers primarily on two key aspects that make them apt for use in biological systems. The first one is the intrinsic properties of the gold core. Secondly, the ability of these particles to target specific cells. Due to their small and spherical shape, their cellular uptake has the best internalization, in addition to effectively delivering drugs by being cationic. In general, due to the overall anionic surface of the cell exterior, cationic nanoparticles can be translocated via favorable electrostatic interactions. The current applications of nanoparticles have focused primarily on the detection of biological molecules or events. This has been accomplished with great success, as discussed in this research and much highlighted recent research by others [19].

Chitosan selected in this research is from an insect specifically, *O. rhinoceros*. Insects consume millions of tons of plant food each year by chewing, nibbling, and sucking it. Insects wreak havoc on all plants. Roots, leaves, flowers, and seeds are all favorites of insects. They may also eat through bark and timber. Although *O. rhinoceros* is a natural decomposer, it is regarded as a common invasive insect of coconut in most areas of the globe, particularly in Southern Asia [20,21], although it is a serious invasive insect of the oil palm (*Elaeis guineesis*) [21]. Because there is a lot of interest in arthropod-derived chitin as an alternative to marine sources, the underutilized chitin from *O. rhinoceros* may help transform pests into money. However, extensive investigations on *O. rhinoceros*’ chitin and its derivatives have yet to be completed [21]. Thus, the purpose of this research is to synthesize and characterize the novel mixture of curcumin-chitosan gold nanoparticles from *O. rhinoceros*’ chitin-mediated synthesis. The successful accomplishments of this research provide scientific validation for chitin and chitosan isolated from *O. rhinoceros* as well as the synthesized CCG-NP that has not yet been reported in the literature.

## 2. Results and Discussion

### 2.1. Characterisation of CCG-NP

Chitosan from *O. rhinoceros* holds great potential in turning pests into wealth. To support the claim stating that chitosan from pests has great value, this research elucidates the commercial potential of the CCG-NP through in vitro drug release profiling and cytotoxicity test. Characterization of CCG-NP was performed by chromatographic and spectrometry techniques as described below.

#### 2.1.1. FTIR

FTIR analysis was carried out to determine the possible biomolecules responsible for the bioreduction, capping, and stabilization of CCG-NPs. Figure 1 shows the FTIR spectra of C2, C10 curcumin, gold (III) chloride, TPP, commercial chitosan, and *O. rhinoceros* chitosan. C2 and C10 were the CCG-NP previously synthesized. Where C2 consisted of 0.5% *O. rhinoceros* chitosan, 60 mM HAuCl_4_, 0.1% curcumin, and 0.1% TPP while C10 had the reaction mixture of 0.5% commercial chitosan, 60 mM HAuCl_4_, 0.1% curcumin, and 0.1% TPP.

The prominent peaks present in CCG-NP for C2 were located at 3321, 2928, 2889, 1661, 1551, 1412, 1015, and 919 cm^−1^. For C10, the prominent peaks present were at 3321, 2948, 2889, 1652, 1603, 1433, 1032, and 972 cm^−1^. Clearly, the spectrum at the absorption bands of 3321 cm^−1^ was related to the stretching vibration of O–H. The absorption bands at 1661–1412 cm^−1^ corresponded to the main chain of chitosan formed by the stretching vibration of N–H and CH_2_. In the case of C2, the absorption bands appeared at 1661 cm^−1^ and 1412 cm^−1^ while C10 peaked at 1652 cm^−1^ and 1433 cm^−1^, which was related to the absorption peaks for O–H, N–H, and CH_2_ groups of chitosan, respectively [22]. The peaks observed at 1015–919 cm^−1^ and 1032–972 cm^−1^ for C2 and C10, respectively, were proof for the presence of a P=O bond of TPP indicating the formation of CCG-NP with TPP. In addition, the peak at 1551 cm^−1^ and 1603 cm^−1^ can be related to the presence of C=C in the aromatic ring and the carbonyl group in the curcumin structure, which is indicative of the successful drug entrapment by the CCG-NP. A similar study was conducted where curcumin was incorporated into chitosan/gold nanogels for enhancing the cytotoxicity activity of curcumin on cancer cell lines. The peaks in the FTIR by the previous study showed very similar absorbance peaks at 3446, 1706, 1600, 1591, 1500, 1392, 1412, and 1068 cm^−1^ [22].

Although it is hard to confirm the presence of specific compounds, the FTIR spectra can still determine the functional groups present in CCG-NP. Based on the peaks, the successful entrapment of curcumin in CCG-NP confirmed the presence of the active ingredients for cosmeceutical applications.

#### 2.1.2. Measurement of Zeta Potential via Zeta Sizer

The greater the value of zeta potential either positive or negative, the more stable the suspension formed will be as compared to lower values of zeta potential. In this study, the zeta potential values were 20.2 ± 3.81, −18.6 ± 3.06, and −16.6 ± 3.27 mV for C2, C10, and C16, respectively (Table 1). C2 showed a positive value. However, C10 and C16 exhibited a negatively charged value. As mentioned previously, the positive and negative charges only show the charged ion present in CCG-NP suspension. C2 had the biggest zeta potential value with 20.2 ± 3.81 mV followed by C10 with −18.6 ± 3.06 mV and lastly C16 with −16.6 ± 3.27 mV. From these results. It was obvious that C2 possessed the highest stability followed by C10 and C16. From the zeta potential of CCG-NPs, it was obvious that C2 possessed the highest stability followed by C10, and lastly C16 as the most unstable mixture. The characteristics of chitosan as a nanocarrier were proven due to the increase of zeta potential for both C2 and C10 as compared to C16. The increased stability of CCG-NP proved the effectiveness of chitosan when incorporated into the nanoparticle suspension [18]. The stability of a solution is substantial in the cosmeceutical industry, especially liquid products. As a result, as the zeta value increases, so does the stability. Hence, chitosan encapsulation of curcumin in gold nanoparticles has proven to be able to maintain the stability of CCG-NP for a longer period [1,23].

#### 2.1.3. Morphological Properties of CCG-NP via FE-SEM

FE-SEM images confirmed the synthesis of CCG-NP. The FE-SEM analysis showed a solid dense spherical structure of CCG-NP with little shape variation where aggregation occurred. From Figure 2a, a more regular spherical shape of C2 can be observed where the sizes were more consistent. This indicated that the surface of CCG-NP was more homogenous and hence showed good compatibility within the mixture.

Similar properties have been found in other nanoparticles with a solid, dense, and homogeneous structure and a spherical shape, according to research. [24,25,26,27]. For C10 (Figure 2b), the shape varied where aggregation and clumping can be seen. The same situation happened in C16 (Figure 2c) but with more aggregation and clumping. This was in accordance with the zeta potential analysis of CCG-NP where the recorded zeta potential values were 20.2 ± 3.81, −18.6 ± 3.06, and −16.6 ± 3.27 mV for C2, C10, and C16, respectively. This situation was proof that chitosan aided in stabilizing the suspension of CCG-NP. In this study, *O. rhinoceros* chitosan (C2) exhibited better stabilizing properties as compared to commercial chitosan (C10) due to having a more regular shape with insignificant aggregation and clumping with higher zeta potential value of 20.2 ± 3.81 mV where it was positively charged. While C10 showed a slightly lower zeta potential value of −18.6 ± 3.06 mV with a negative charge. However, both *O. rhinoceros* chitosan and commercial chitosan showed better stabilizing properties as compared to the mixture of CCG-NP without chitosan. The zeta potential measurement of C16 showed a much lower value of −16.6 ± 3.27 mV with a negative charge. The zeta potential analysis supported the morphological study of CCG-NP by FE-SEM.

The size of nanoparticles ranges from 5 nm to 400 nm [1]. The interparticle interactions between curcumin and chitosan toward gold nanoparticles’ assembly are the ones making up their characteristics. The assembly usually comes in a variety of shapes, and in this research, the shape is in nanospheres. Because of their small size and thus resulting in having a huge surface area, a spherical shape, and a high zeta potential value, they have a high drug-loading capacity and can easily reach their target cell [1].

#### 2.1.4. Particle Size Distribution Analysis via Zeta Sizer

The average particle sizes were 128.27 ± 0.00, 171.96 ± 4, 339.66 ± 11.36, and 465.74 ± 0.00 nm for C2, C10, C16, and 0.1% curcumin in methanol, respectively. There was a significant decrease in the average particle size. Here, 0.1% curcumin had the biggest average particle size followed by C16. C10 and C2 showed significantly smaller sizes compared to 0.1% curcumin and C16. A drop of 328.47 nm in average size could be seen for C2 where the particle size of curcumin being stabilized in chitosan-gold nanoparticle was from 465.74 ± 0.00 nm to 128.27 ± 0.00 nm. Meanwhile, for C10, a drop of 293.78 nm could be seen in the average particle size of the mixture. A decrement in C10 could be seen from 465.74 ± 0.00 nm to 171.96 ± 4 nm. In C16, without chitosan, the decrement of average particle size still happened but with much less drop as compared to when the mixture was being stabilized with the chitosan-gold nanoparticle. The reduction was only 126.08 nm as compared to C2 and C10 with 328.47 nm and 293.78 nm, respectively. It was a decline from 465.74 ± 0.00 nm to 339.66 ± 11.36 nm. The average particle sizes of CCG-NP were in accordance with the morphology study as well as the zeta potential measurement. The images from FE-SEM showed an increase in size and irregularity in shape as the average particle size increased. The zeta potential value also decreased with the increase of average particle size indicating a less stable suspension of CCG-NP and vice versa.

#### 2.1.5. Crystallinity via X-ray Diffraction (XRD)

XRD analysis was carried out to confirm the crystalline nature of the synthesized CCG-NP. The XRD analysis showed an amorphous state for CCG-NP. The XRD diffractogram of curcumin showed multiple peaks between 5 and 30° [28], which were mainly attributed to its crystalline nature. These characteristic peaks had disappeared in the CCG-NP (Figure 3) suggesting the transformation of the crystalline nature of curcumin to an amorphous state [29]. This change in physical characteristics may perhaps result from the molecular interactions between curcumin and chitosan occurring during formulation. A similar XRD pattern of curcumin-polylactic-co-glycolic acid nanoparticles was reported [30]. It is worthwhile to note that the physical transformation of curcumin in the CCG-NP did not affect the structural characteristics of curcumin, as confirmed by the FTIR analysis. A similar report was found where the previous XRD peaks at 19.7° due to its crystalline nature disappeared, suggesting the transformation of the crystalline nature of curcumin, and chitosan as an individual changed to an amorphous state [28,29].

#### 2.1.6. EDX

EDX spectra (Figure 4) confirming the elemental composition of CCG-NP as indicated by the strong signals for an elemental gold presence in C2, C10, and C16 are 52.38%, 51.92%, and 52.39% in weight. Apart from the EDX signal for elemental gold, all the EDX spectra also showed signals for C and O, which were most probably from the bioreductants present in the mixture acting as coating and stabilization for CCG-NP [29]. The observed elemental composition in CCG-NP for C and O were 1.98% and 26.15% for C2; 2.17% and 25.46% for C10; and 1.79% and 25.25% for C16, respectively. Cl was from the gold (III) chloride of the gold nanoparticle as the reactive compound in the mixture of CCG-NP. In each respective CCG-NP, the percentage composition for Cl was very small with 0.42%, 0.01%, and 0.03% for C2, C10, and C16. While the additional weak signal of Si was probably present due to the glass slide used for gold sputter coating during the preparation of the sample for EDX analysis. A similar EDX profile was reported previously where the observed elemental composition in chitosan encapsulated nanoparticles using TPP as crosslinker (32.9% C, 48.4% O, Na 9.9%, 5.2% N, and 3.6% P) and curcumin nanoparticles with TPP as crosslinkers (36.2% C, 47.8% O, Na 8.0%, 5.1% N, and 2.9% P) [29]. Gold or Au plays a critical role in the synthesis of CCG-NP. Therefore, it is necessary to ascertain the existence of gold in CCG-NPs. As a result, the presence of gold determines the quality of gold nanoparticles in the cosmeceutical sector. Many cosmeceutical companies claim that their products include gold nanoparticles. However, there is no way of knowing the presence of actual gold in the product without proper conduct of analysis. As a result, the presence of gold from elemental analysis in this research proved the product’s authenticity [23].

#### 2.1.7. Toxicity Test by MTT Assay

The activity is reported as IC50 value, which are the concentrations of CCG-NP required to exhibit 50% cell viability of 3T3 cells. Results in Figure 5 demonstrated the value of IC50 (50% inhibition) of CCG-NP on 3T3 cells after 72 h of exposure occurs at 58%. Here, 1% of CCG-NP exhibited 74% of cell viability and the value decreased as the concentration of CCG-NP increased where 90% of CCG-NP resulted in 42% of cell viability. It has been reported that a low level of curcumin between 1–5 µg/mL demonstrated no cytotoxicity [31].

According to ISO 10993-5:2009 (E), biological evaluation in vitro tests for cytotoxicity, when there is no reduction or cell death of less than 20%, there is only slight cytotoxicity. Meanwhile, 50% of cell death is considered to have a mild cytotoxic effect, which is still in the acceptable zone. More than 70% of cell death is considered moderate cytotoxicity, while nearly complete destruction of cell layers is considered severe cytotoxicity. Therefore, IC50 of 58% for CCG-NP is considered mild cytotoxicity. Based on similar findings from previous research, the presence of chitosan-chromone derivative at concentrations up to 800 µg/mL had no influence on the cellular viability of cultured mouse embryonic fibroblast (MEF) cells, as determined by cytoplasmic esterase enzyme activity and plasma membrane integrity [32]. The chitosan-chromone derivative-treated cells grew at the same rate as the untreated control cells. Kumar and Koh (2012) demonstrated unequivocally that the chitosan-chromone derivative was non-cytotoxic and had no influence on cellular growth [32]. Therefore, the MTT assay was used to evaluate the safe usage dose and, as a result, CCG-NP is considered to be a mild-to-moderate toxicity mixture for consumers. With a controlled usage dose, CCG-NP is safe for topical application in cosmeceutical products [1].

#### 2.1.8. Significance of the Applied Value of This Research in Cosmeceutical Application

The cosmetic industry aims at developing an environmentally friendly nanoparticulate system with high therapeutic benefits for the skin. Hence, the non-toxic, nanosized chitosan drug carrier is very promising in cosmeceutical applications. This CCG-NP was able to have the chitosan encapsulate and stabilize curcumin in gold nanoparticles. The encapsulated curcumin was protected from degradation, thus increasing its bioavailability. The toxicity evaluation also proved that CCG-NP is safe for use. Therefore, the introduction and exploration of new materials from the *O. rhinoceros* beetle represent an innovative natural compound that can certainly be utilized in cosmeceutical applications as a potent skin whitening agent.

## 3. Material and Methods

### 3.1. Preparation of Chitosan from O. rhinoceros

For chitin extraction deproteinization was carried out by a strong base (1 M NaOH, 99 °C, 20 *v/w* of sample) [33], while decalcification was carried out by a strong acid (3.9 M HCl, 75 °C, 12 *v/w* of sample) [34] with few modifications. The process continued with the deacetylation of chitin where 10 g of chitin was put in 200 mL of 60% NaOH at 99 °C for 60 min. The solution was then filtered and washed until pH 7. The mush was oven-dried overnight at 40 °C. The dried sample was ground using mortar and pestle. Afterward, the ground sample was sieved using a 500 µm sieve and kept in a 4 °C airtight container for further use.

### 3.2. Preparation of CCG-NP

The preparation of CCG-NP was based on a previously reported method [28,35,36] with a few modifications. CCG-NP was prepared with a volume ratio of 2:0.5 by adding 2 mL 60 mM HAuCl_4_ solution to 0.5 mL of 0.1% curcumin solution in a clear universal bottle covered with aluminum foil. To synthesize the curcumin loaded with chitosan into gold nanoparticles, an aqueous solution of TPP (0.1%) was mixed with chitosan solution (3 mL, 0.5%) while magnetic stirring the solution. After 50 min, an aqueous solution of HAuCl_4_ (2 mL, 60 mM) was added, and the mixture was heated to 70 °C [37] and stirred for another 45 min [22]. The bottle was left for 24 h at room temperature. The same method was applied for C2, C10, and C16 with variables being chitosan from *O. rhinoceros*, commercial grade chitosan, and blank, respectively.

### 3.3. Characterization of CCG-NP

#### 3.3.1. Fourier Transform Infrared (FTIR) Spectroscopy

FTIR spectra were obtained by using Fourier transform infrared spectrometer (FT-IR-ATR spectrometer SPECTRUM 2, PerkinElmer, Buckinghamshire, UK). 10 µL of CCG-NP solution was placed on the FTIR grid for obtaining the FTIR spectra. FTIR in the range of 4000–450 cm^−1^ [38].

#### 3.3.2. Measurement of Zeta Potential via Zeta Sizer

The ζ-potential values of CCG-NP and as-synthesized colloidal CCG-NP were determined by Zetasizer (Malvern Instruments, Malvern, UK). For this, 1.5 mL of the synthesized CCG-NP solution was put in a folded capillary zeta cell and 12 zeta runs were performed at 25 °C [39].

#### 3.3.3. Morphological Properties of CCG-NP via FE-SEM

The surface morphology of CCG-NP was examined with a field emission scanning electron microscope, (FE-SEM, JEOL JSM-7600F, Hitachi, Tokyo, Japan). The liquid samples were diluted with distilled water. A few droplets of the diluted samples were dropped on a clean glass slide followed by drying in a hot air oven at 40 °C to obtain a thin film. The glass slide with the dried samples was fixed on adhesive tape and then coated with a thin gold layer by a sputter coater. The FE-SEM was conducted accordingly. The dried CCG-NP on the glass slide was finally gold-sputtered on JCF-1600 Auto Fine Coater. The gold-sputtered glass slide was then mounted on the specimen stub with double-sided adhesive tape and examined under FESEM at an accelerating voltage of 10 keV [40].

#### 3.3.4. Particle Size Distribution Analysis via Zeta Sizer

The particle size of CCG-NP was measured at 25 °C using Zetasizer (Malvern Instruments, Malvern, UK). The particle size was determined using the dynamic light scattering method, with scattered light collected at 173 °C. The Z-Ave value was reported as the mean diameter of CCG-NP when data were analyzed using the cumulant value [41].

#### 3.3.5. Crystallinity via X-ray Diffraction (XRD)

The sample was oven-dried overnight and was carefully put on a glass slide and placed in the specimen holder and powder X-Ray diffraction measurements were carried out on Shimadzu XRD 6000 diffractometer. Cu-Kα1 radiation with λ of 1.5406 Å was used to record the XRD pattern with a nickel monochromator filtering the wave at a tube voltage of 40 kV and tube current of 30 mA. The scanning was carried out in the region of 2θ from 30° to 80° at a speed of 0.02°/min and the time constant was 2 s [42].

#### 3.3.6. Elemental Analysis via Energy Dispersion X-ray (EDX)

The EDX spectra were used for the elemental analysis and quantitative determination of available elements on the surface of CAP-GNPs synthesized under different conditions. The EDX analysis was performed on FESEM (JEOL JSM-7600F, Hitachi, Tokyo, Japan) having an EDX detector operating at the accelerating voltage of 20 keV [43].

#### 3.3.7. Toxicity Test by MTT Assay

In this research, the parameters used to determine cytotoxicity is by using MTT assay (3-[4,5-dimethylthiazol-2-yl]-2,5-diphenyltetrazolium bromide). The 3T3 cells used were from human epithelial colorectal adenocarcinoma cell line. Cell culture with the concentration of 2 × 10^3^ cells/mL was prepared and plated (100 µL/well) onto 96-well plates [42].

## 4. Conclusions

Since it is substantial to have an effective cosmeceutical product, incorporating curcumin into cosmetics formulation will certainly turn it into an effective therapeutic product. Curcumin is an active compound with many benefits. To achieve the goal of cosmeceuticals in dermatology research, it is important to produce new ecologically friendly nanosized compounds for skin nanoparticulate systems. All characterization involves comparing CCG-NP with *O. rhinoceros* chitosan to commercial chitosan with the same exact conditions. Based on the analyses of FTIR, XRD, zeta potential, and chitosan encapsulation of curcumin in gold nanoparticles, respectively, it is concluded that the *O. rhinoceros*-extracted chitosan has demonstrated similar performance to commercial chitosan. The synthesized CCG-NP from *O. rhinoceros* chitosan is a stable, spherical, nano-sized, non-toxic, and homogeneous solution and hence has a very high potential in cosmeceutical applications.

## Figures and Tables

**Figure 1 molecules-28-01799-f001:**
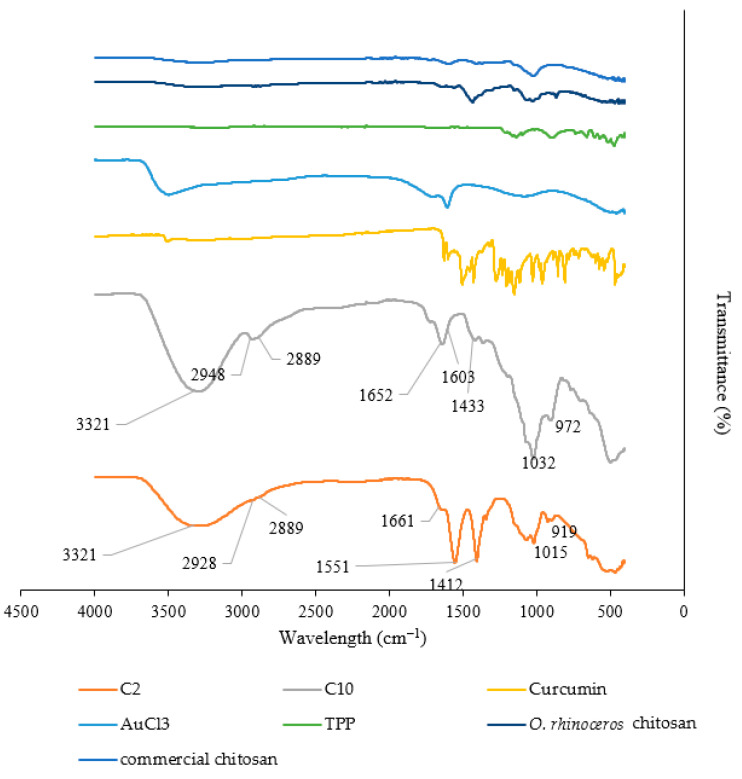
FTIR spectra of C2, C10, curcumin, gold (III) chloride, TPP, *O. rhinoceros* chitosan, and commercial chitosan.

**Figure 2 molecules-28-01799-f002:**
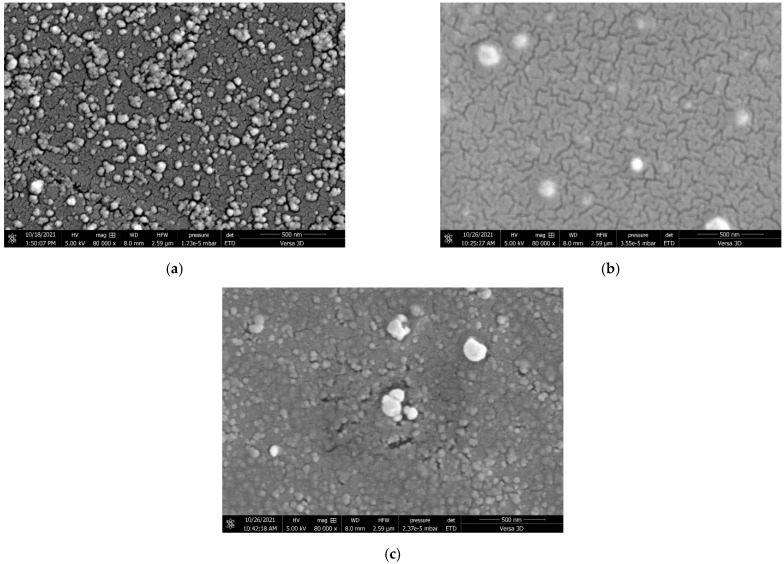
FE-SEM images showing the spherical morphology of CCG-NP (**a**) C2, (**b**) C10, and (**c**) C16 at 80,000× magnification.

**Figure 3 molecules-28-01799-f003:**
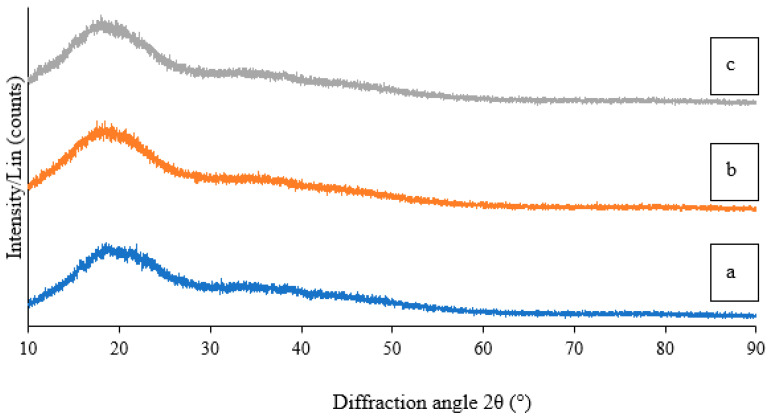
The image shown is an XRD spectrum of (**a**) C2, (**b**) C10, and (**c**) C16 in their amorphous state.

**Figure 4 molecules-28-01799-f004:**
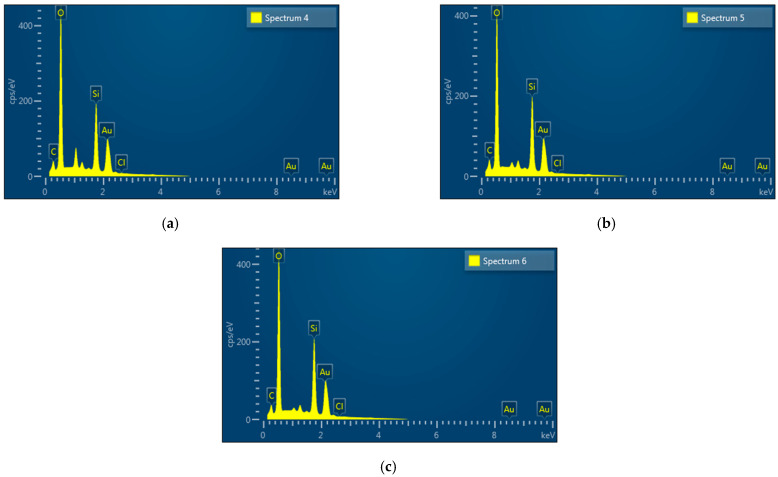
EDX spectra of CCG-NP (**a**) C2, (**b**) C10, (**c**) C16.

**Figure 5 molecules-28-01799-f005:**
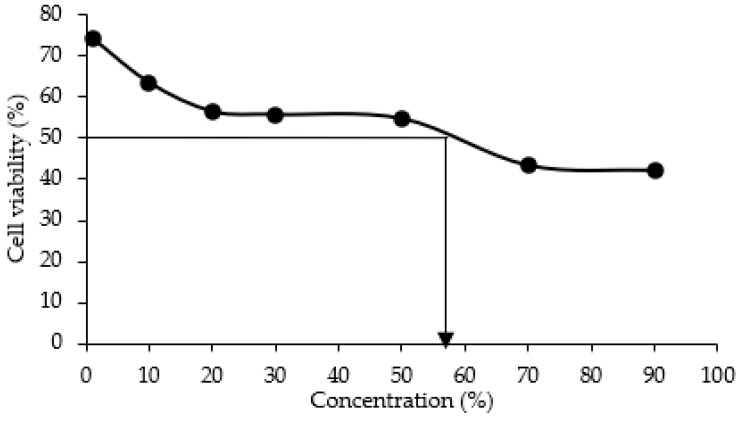
Effect of CCG-NP on 3T3 cells after 72 h of exposure.

**Table 1 molecules-28-01799-t001:** Zeta potential of CCG-NP; mean ± S.D. (*n* = 3).

CCG-NP	Zeta Potential ± sd (mV)
C2	20.2 ± 3.81
C10	−18.6 ± 3.06
C16	−16.6 ± 3.27

## Data Availability

Not applicable.

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
