# Peer review of "Synthesis and Characterization of Curcumin-Chitosan Loaded Gold Nanoparticles by Oryctes rhinoceros’ Chitin for Cosmeceutical Application"

_molecules, 2023, doi:10.3390/molecules28041799_

Round 1

Reviewer 1 Report

This paper reports "Synthesis and Characterization of Curcumin-Chitosan Loaded 2 Gold Nanoparticles by Oryctes rhinoceros’ Chitin for Cos- 3 meceutical Application". However, I cannot find any property result. Futhermore, the characterization is lack, e.g. TEM images. Overall, this paper is not outstanding in terms of novelty and research work. I suggest the authors provide enough data and resubmit it. 

Author Response

Response 1: The primary goal of this work is to investigate the chitin and chitosan extracted from O. rhinoceros as well as the unpublished CCG-NP that was created in the lab. The majority of research work, according to peer-reviewed journal articles we've studied, typically focuses on the characterization of nanoparticles. Examples of this include using FTIR, a zeta sizer to measure zeta potential, particle size distribution analysis, FE-SEM, EDX, and X-ray diffraction (XRD). We have improved the results and discussion in response to the reviewer's concerns, as advised by the esteemed reviewer. We appreciate the respected reviewer's comments on the TEM analysis, however, our institutes are not equipped to characterize CCG-NP using TEM. As a result, we wish to help the student realize the limitations of the sources that are available.

Reviewer 2 Report

There are the following comments, which is needed to be addressed by the authors.

1) Why did authors used Au in preparing the formulations. What is the benefits? 

2) Authors should write in detail the preparation of C2, C10 and C16.

3) SEM images show only CCP NPs. Authors should show the presence of Au.

4) How much curcumin was loaded in CCP NPs?

5) Authors should present zeta potential data in the form of table.

6) Authors need to show the EDX spectra.

7) Authors should discuss how these formulations could be useful in cosmetic applications.

8) Authors should study the dissolution of CCP NPs in physiological conditions.

Author Response

Point 1: Why did authors used Au in preparing the formulations. What is the benefits?

Response 1:  The active application of AuNPs center primarily on two key aspects that make them apt for use in biological systems. The first one being the intrinsic properties of the gold core. Secondly, the ability of these particles to target specific cells. Due to their small and spherical shape, their cellular uptake has the best internalization, in addition, to effectively delivering drugs by being cationic. In general, due to the overall anionic surface of the cell exterior, cationic nanoparticles are able to be translocated via favorable electrostatic interactions. The current applications of nanoparticles have focused primarily on the detection of biological molecules or events. This has been accomplished with great success, as discussed in this research and many highlighted recent researches by others.

Point 2: Authors should write in detail the preparation of C2, C10 and C16.

Response 2:  The preparation of C2, C10 and C16 was discussed more in ‘Preparation of CCG-NP’

Point 3: SEM images show only CCP NPs. Authors should show the presence of Au.

Response 3:  The presence of Au was proven in EDX

Point 4: How much curcumin was loaded in CCP NPs?

Response 4:  The concentration of curcumin loaded in CCG-NP was stated in ‘2.2 Preparation of CCG-NP’ which is 0.1% curcumin in methanol with 2:0.5 ratio of gold against curcumin.

Point 5: Authors should present zeta potential data in the form of table.

Response 5: Data added and presented in table 1

Point 6: Authors need to show the EDX spectra.

Response 6: EDX spectra added and shown in Figure 4

Point 7: Authors should discuss how these formulations could be useful in cosmetic applications.

Response 7: Added and discussed in “3.1.8. Significance of the applied value of this research in cosmeceutical application.”

Point 8: Authors should study the dissolution of CCP NPs in physiological conditions.

Response 8: The toxicity of CCG-NP was added to the manuscript.

Reviewer 3 Report

The authors described the preparation of chitosan from O. rhinoceros chitin, the synthesis of curcumin loaded chitosan-gold nanoparticles, and particularly, the characterization of the nanoparticles. It is an interesting work showing a potential value-increment  way to use crustacean chitin.

As the authors mentioned in the title of the manuscript, the nanoparticles were synthesized for cosmeceutical applications, therefore, preliminary data regarding the biological values, the toxicity of the nanoparticles should be provided. And a comparison of the cosmeceutical functions among curcumin, gold nanoparticle and the nanoparticles studied in this manuscript will be of value for readers to understand the importance of this work.

Author Response

Point 1: The authors described the preparation of chitosan from O. rhinoceros chitin, the synthesis of curcumin loaded chitosan-gold nanoparticles, and particularly, the characterization of the nanoparticles. It is an interesting work showing a potential value-increment  way to use crustacean chitin.

As the authors mentioned in the title of the manuscript, the nanoparticles were synthesized for cosmeceutical applications, therefore, preliminary data regarding the biological values, the toxicity of the nanoparticles should be provided. And a comparison of the cosmeceutical functions among curcumin, gold nanoparticle and the nanoparticles studied in this manuscript will be of value for readers to understand the importance of this work.

Response: Few alterations were made to the manuscript. The toxicity of CCG-NP was added to the manuscript as per suggestion. 

Round 2

Reviewer 1 Report

The authors addressed the reviewer's commnets. I suggest this paper can be accepted.

Author Response

Thank you so much for your kind review. We really appreciate you taking the time to share your experience with us. 

Reviewer 3 Report

The authors provided cell viability results to show the safety of the CCG-nanoparticles. I think the manuscript can be published by "Molecules". 

In addition, I have some minor suggestions. 1) In the 3.1.8 section, the authors mentioned the CCG-Nanoparticles inhibited E. coli and S. aureus. If the authors performed the experiments, it is better to add the data in the manuscript. Otherwise, the auhors should provide the references and describe clearly, for example, the concentrations of CCG-Nanoparticles used. 2) In line 321 of the revised manuscript, the authors mentioned a concentration of "800 g/mL" chitosan-chromone derivative, really??

Author Response

Response 1: The author sent a manuscript to publish antimicrobial and anti-tyrosinase data elsewhere. However, the manuscript has yet to be published. Hence the author decides to remove these two statements.

Response 2: Thank you for the comment. I have checked the value and corrected the unit.